# A fluorescence anisotropy assay to discover and characterize ligands targeting the maytansine site of tubulin

Grégory Menchon[1], Andrea E. Prota[1], Daniel Lucena-Agell[2], Pascal Bucher[3], Rolf Jansen[4], Herbert Irschik[4], Rolf Müller[5], Ian Paterson[6], J. Fernando Díaz [2], Karl-Heinz Altmann[3] & Michel O. Steinmetz [1,7]

Microtubule-targeting agents (MTAs) like taxol and vinblastine are among the most successful chemotherapeutic drugs against cancer. Here, we describe a fluorescence anisotropy-based assay that specifically probes for ligands targeting the recently discovered maytansine site of tubulin. Using this assay, we have determined the dissociation constants of known maytansine site ligands, including the pharmacologically active degradation product of the clinical antibody-drug conjugate trastuzumab emtansine. In addition, we discovered that the two natural products spongistatin-1 and disorazole Z with established cellular potency bind to the maytansine site on β-tubulin. The high-resolution crystal structures of spongistatin-1 and disorazole Z in complex with tubulin allowed the definition of an additional sub-site adjacent to the pocket shared by all maytansine-site ligands, which could be exploitable as a distinct, separate target site for small molecules. Our study provides a basis for the discovery and development of next-generation MTAs for the treatment of cancer.

[1] Laboratory of Biomolecular Research, Division of Biology and Chemistry, Paul Scherrer Institut, Villigen PSI 5232, Switzerland. [2] Chemical and Physical Biology, Centro de Investigaciones Biológicas, Consejo Superior de Investigaciones Cientificas CIB–CSIC, Madrid 28040, Spain. [3] Department of Chemistry and Applied Biosciences, Institute of Pharmaceutical Sciences, ETH Zürich, Zürich 8093, Switzerland. [4] Abteilung Mikrobielle Wirkstoffe, Helmholtz Zentrum für Infektionsforschung, Braunschweig 38124, Germany. [5] Department Microbial Natural Products and Department of Pharmacy at Saarland University, Helmholtz Institute for Pharmaceutical Research Saarland, Helmholtz Centre for Infection Research, Saarbrücken 66123, Germany. [6] University Chemical Laboratory, Cambridge University, Cambridge CB2 1EW, UK. [7] University of Basel, Biozentrum, Basel 4056, Switzerland. These authors contributed equally: Andrea E. Prota, Daniel Lucena Agell, Pascal Bucher.  Correspondence and requests for materials should be addressed to M.O.S. (email: michel.steinmetz@psi.ch)

The αβ-tubulin heterodimer is the building block of microtubules that, together with F-actin and intermediate filaments, constitute the cytoskeleton. As such, tubulin is an important target for antineoplastic drugs like taxol and vinblastine[1]. By perturbing microtubule dynamics during mitosis, these drugs interfere with mitotic spindle formation and thus cell division; however, they also act on interphase microtubules and, as a consequence, affect the intracellular trafficking of important molecules and organelles, in particular in neurons[2,3].

Microtubule-targeting agents (MTAs) can be broadly divided into microtubule-stabilizing and -destabilizing agents. Six distinct tubulin-binding sites for ligands have been structurally characterized to date, which are referred to as the taxane, laulimalide/peloruside, colchicine, vinca, pironetin and maytansine site, respectively[4–7]. Molecules that bind to the taxane and laulimalide/peloruside site stabilize microtubules, while compounds targeting the colchicine, vinblastine, pironetin or maytansine site destabilize microtubules. Taxane- and vinblastine-site ligands are in clinical use for cancer therapy, but no drugs have been approved that target any of the other four binding sites, with the exception of maytansine that is part of an antibody-drug conjugate (ADC). Furthermore, the clinical application of approved MTAs is hampered by their severe toxic side effects and the development of resistance[8].

The maytansine site on tubulin has been discovered only very recently[9]. It is a unique site on β-tubulin that is located at the longitudinal tubulin–tubulin interface in microtubules, which readily explains the microtubule-destabilizing effects of maytansine-site ligands. Three distinctly different ligands that target the maytansine site have been described: maytansine, PM060184 and rhizoxin[9]. Maytansine has been successfully incorporated into the ADC trastuzumab emtansine, which is approved for the treatment of breast cancer[10]. In principle, ADCs can overcome the toxicity problem associated with MTAs; however, the costs for developing and using ADCs in targeted therapy are a major drawback[11] and the use of traditional anti-tubulin agents still remains a valuable approach. PM060184 is in phase II clinical development for the treatment of breast cancers (clinicaltrials.gov), while rhizoxin had reached phase II, before being discontinued for reasons that are poorly understood[12].

In this study, we develop a quantitative fluorescence anisotropy displacement assay based on a fluorescein-labeled maytansine derivative, with the aim to provide a platform for the identification and characterization of additional maytansine-site ligands. We choose the maytansine site because it is poorly characterized and because no tools are available to characterize the binding of maytansine-site ligands in detail. We show that the assay is specific for the maytansine site and can be operated in a high-throughput manner. By using this assay, we identify two natural products, spongistatin-1 and disorazole Z, as maytansine-site ligands. We solve the structures of both compounds in complex with tubulin to high resolution by X-ray crystallography, which allows us to analyze the maytansine site in great detail. The experimental tools and results presented in this study should contribute to the discovery and characterization of maytansine site-directed, small-molecule MTAs for the development of next-generation anti-tubulin drugs for the treatment of cancer.

## Results

**A fluorescent probe targeting the maytansine site of tubulin**. In this study, we sought to develop a fluorescence anisotropy assay to identify and determine the tubulin-binding affinities of maytansine-site ligands. To this end, we prepared a fluorescently labeled maytansinoid that carries a fluorescein reporter attached to the 3-OH group of the maytansinol core structure via a flexible linker moiety (referred to as FcMaytansine (**M5**)). In the tubulin–maytansine crystal structure[9], the corresponding N-acetyl-N-methyl-D-alanyl residue in maytansine occupies a solvent-exposed position that is located remote from the tubulin surface; thus, the fluorescein-linker moiety was expected not to interfere with binding of the FcMaytansine probe. As shown in Fig. 1, FcMaytansine was prepared in two steps from commercially available maytansinol (**M1**). Details of the synthesis can be found in the Supplementary Information.

The tubulin-binding properties of FcMaytansine were assessed by fluorescence anisotropy measurements. Using our experimental setup (see Methods), the addition of increasing amounts of tubulin (0 to 400 nM) to FcMaytansine (10 nM) led to a concentration-dependent increase in anisotropy by a factor of ~1.6 from ~16 mA in the absence of tubulin to ~250 mA at saturation (Fig. 2a), thus providing an assay with both excellent dynamic range and signal-to-noise ratio (see also Supplementary Fig. 1a). We observed that the increase in fluorescence anisotropy was accompanied by a concomitant increase in the fluorescence

**Fig. 1** Synthesis of FcMaytansine. **a** M2, Et₃N, 4-(1-pyrrolidinyl)-pyridine, rt, 7d, then prep. RP-HPLC, 20%; **b** M4, CuSO₄, Na-ascorbate, THF/H₂O 4:1, 5 h, then prep. RP-HPLC, 29%

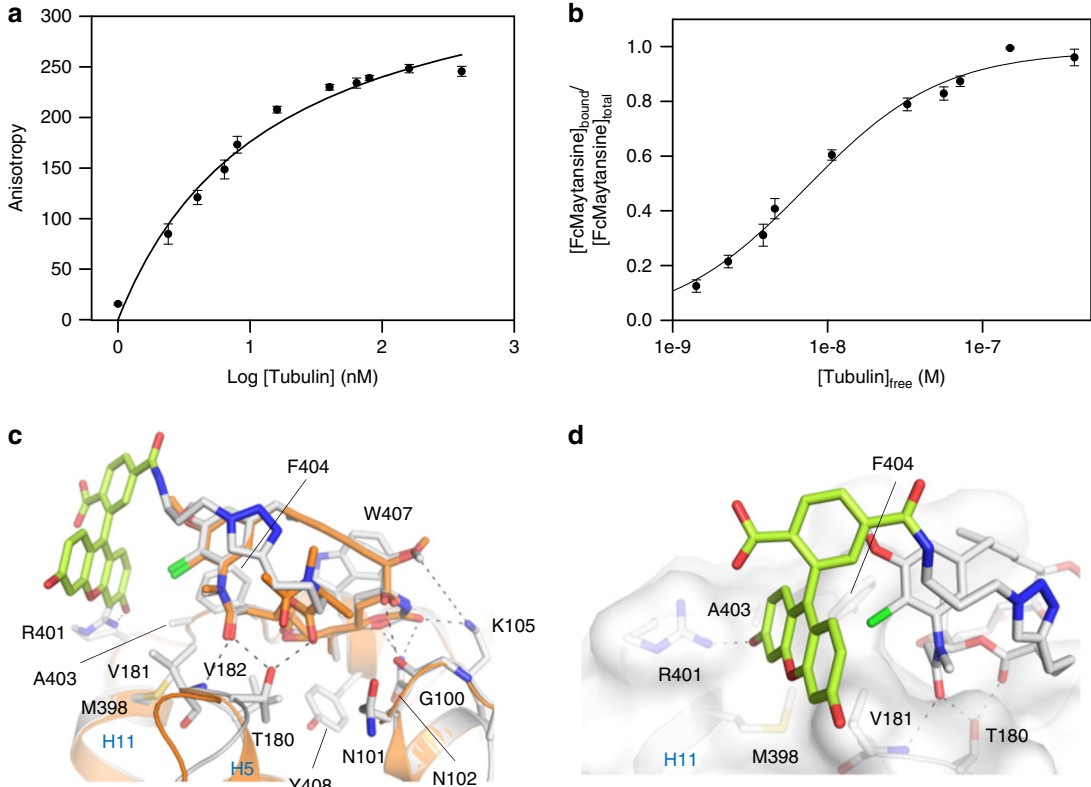

**Fig. 2** Binding of FcMaytansine to tubulin. **a**, **b** FcMaytansine (10 nM) anisotropy variation (**a**) and fraction bound (**b**) as a function of the tubulin concentration. The data are from three independent experiments and represent mean ± SEM. The solid lines represent fits to the data (see Methods). **c** Crystal structure of the tubulin–FcMaytansine complex. Superposition of FcMaytansine (white/green) and maytansine (orange, PDB ID 4TV8) bound to tubulin. The superimposed β-tubulin chains are in ribbon, both the ligands and the interacting residues are in stick representation and are labeled. **d** Close-up view of the interactions formed by the xanthene-moiety of FcMaytansine. The same display settings as in **c** are used

intensity by a factor of 3.1 ± 0.1. This behavior is well known for fluorescein derivatives, whose fluorescence properties depend on the exact pKa value of the carboxyl group, which is often affected upon binding of a fluorescein-labeled probe to a protein target[13,14]. Incubation of samples for 15 and 45 min at 25 °C prior to measurements resulted in the same anisotropy values, demonstrating that a pre-incubation time of 15 min was sufficient for the system to equilibrate (Supplementary Fig. 1b). The fluorescence anisotropy data were fitted using a single binding site model, which yielded a dissociation constant, $K_d$, for the tubulin–FcMaytansine interaction of 6.8 ± 0.8 nM (mean ± SEM; $n = 5$) (Fig. 1b). A $K_d$ ~of 900 nM has been reported previously for the tubulin–maytansine interaction[15]; the ~60-fold lower $K_d$ value that we obtained for FcMaytansine might be due to the different assay type and conditions used in our study.

To demonstrate that FcMaytansine does indeed specifically bind to the maytansine site of tubulin, we used X-ray crystallography. FcMaytansine was soaked into a crystal composed of a complex formed between two αβ-tubulin heterodimers ($T_2$), the stathmin-like protein RB3 (R) and tubulin tyrosine ligase (TTL)[16,17], and the structure of the $T_2$R–TTL–FcMaytansine complex was solved at 2.4 Å resolution (Table 1). As shown in Fig. 2c, d, the structure revealed that the maytansinoid moiety of FcMaytansine interacts with the maytansine site of β-tubulin in exactly the same manner as the parent compound maytansine[9]. The 3'-carboxyl group of the fluorescein moiety of FcMaytansine loosely packs against the tubulin surface that is formed by the side chains of βVal181, βMet398, βArg401, βAla403 and βPhe404 and establishes a polar contact with the side chain of βArg401. Moreover, the fluorescein moiety is oriented such as to allow the

11-chloro substituent of the maytansine core to establish an edge-on Cl–π interaction with the xanthene ring; this finding may explain the change in fluorescence intensity observed upon binding of the FcMaytansine molecule to tubulin (see above).

Next, we performed fluorescence anisotropy competition experiments with FcMaytansine and the two unlabeled maytansine-site ligands ansamitocin P3 (a maytansine derivative) and PM060184[18,19]. Vinblastine and colchicine were used as negative controls. The concentrations of FcMaytansine (10 nM) and tubulin (4 nM) were chosen such that the bound fraction of FcMaytansine was around 0.5 based on the $K_d$ of the tubulin–FcMaytansine complex. The mixture was then incubated with an unlabeled competitor at increasing concentrations prior to anisotropy measurements. The final dimethyl sulfoxide (DMSO) concentration in each sample amounted to ~1.2%; as documented in Supplementary Fig. 1c, the assay tolerates a DMSO concentration of up to 5%. The plots shown in Fig. 3a, b demonstrate that FcMaytansine can be fully displaced by the two maytansine-site ligands ansamitocin P3 and PM060184 in a dose-dependent manner, but not by vinblastine or colchicine. The $K_d$ values determined for ansamitocin P3 and PM060184 were 15.0 ± 1.3 nM (mean ± SEM; $n = 3$) and 111 ± 30 nM (mean ± SEM; $n = 3$), respectively.

To assess whether our assay is suitable for high-throughput screening, we determined its Z-factor[20]. Samples of 10 nM FcMaytansine mixed with 400 nM tubulin and of 10 nM FcMaytansine alone were applied to a 96-well plate and the anisotropy was measured (Supplementary Fig. 1d). A Z-factor of 0.97 was calculated from the mean and the standard deviation anisotropy values, which underpins the excellent

**Table 1 Data collection and refinement statistics**

|  | T$_2$R–TTL–Disorazole Z | T$_2$R–TTL–Spongistatin-1 | T$_2$R–TTL–FcMaytansine |
|---|---|---|---|
| **Data collection** |  |  |  |
| Space group | P2$_1$2$_1$2$_1$ | P2$_1$2$_1$2$_1$ | P2$_1$2$_1$2$_1$ |
| Cell dimensions |  |  |  |
| a, b, c (Å) | 104.4, 157.6, 179.6 | 105.7, 159.9, 181.0 | 104.4, 156.7, 181.0 |
| α, β, γ (°) | 90, 90, 90 | 90, 90, 90 | 90, 90, 90 |
| Resolution (Å) | 56.0–2.1 (2.15–2.10) | 50.2–2.4 (2.49–2.40) | 49.6–2.4 (2.46–2.40) |
| $R_{merge}$ | 10.5 (312.6) | 16.7 (273.4) | 21.6 (258.6) |
| I/σI | 18.5 (0.7) | 15.8 (1.1) | 10.0 (0.8) |
| Completeness (%) | 100 (99.8) | 99.8 (97.5) | 100 (99.8) |
| Redundancy | 12.1 (7.9) | 16.7 (15.6) | 6.8 (6.5) |
| **Refinement** |  |  |  |
| Resolution (Å) | 56.0–2.1 | 48.1–2.4 | 49.6–2.4 |
| No. of reflections | 172,624 | 115,478 | 116,054 |
| $R_{work}$/$R_{free}$ | 17.9/21.3 | 16.8/21.6 | 20.1/25.8 |
| No. of atoms |  |  |  |
| Protein | 17,632 | 17,648 | 17,542 |
| Ligand | 100 | 180 | 145 |
| Water | 864 | 572 | 453 |
| B-factors |  |  |  |
| Protein | 64.8 | 71.5 | 62.6 |
| Ligand | 115.0 | 112.4 | 97.5 |
| Water | 63.9 | 67.7 | 53.8 |
| R.m.s. deviations |  |  |  |
| Bond lengths (Å) | 0.004 | 0.004 | 0.002 |
| Bond angles (°) | 0.621 | 0.677 | 0.545 |

*For each structure data were collected from a single crystal
*Values in parentheses are for highest-resolution shell

performance of the assay and makes it suitable for high-throughput screening[20].

Together, these results demonstrate that FcMaytansine is a specific probe that enables to distinguish between ligands binding the maytansine site or the nearby vinblastine site of tubulin (Supplementary Fig. 2a). They further establish our fluorescence anisotropy assay as a powerful, high-throughput qualified tool to discover and characterize additional maytansine-site ligands.

**Analysis of trastuzumab emtansine degradation products.** The maytansine derivative DM1 (Fig. 3c) forms the basis for trastuzumab emtansine, one of the two ADCs currently in clinical use for cancer treatment; specifically, trastuzumab emtansine is approved for the treatment of HER2-positive metastatic breast cancer[10]. In cells, trastuzumab emtansine is ultimately transformed into the lysine conjugate **M9** (Fig. 3c) as the pharmacologically active entity by lysosomal degradation of the targeting antibody carrier[10,21]. The in vitro cytotoxicity of **M9** has been investigated and the compound is ~450-fold less potent in cellular assays than the simple methyl thioether of DM1 (S-Me DM1), most likely as a consequence of reduced cellular uptake[22]. At the same time, the intracellular formation of **M9** from an anti-EpCAM-DM1 conjugate in MCF-7 cells has been shown to correlate with the effects of the conjugate on microtubule dynamicity[23]. While this finding indicates that the cytotoxicity of the anti-EpCAM-DM1 conjugate is based on **M9**-mediated suppression of microtubule dynamics, to the best of our knowledge, a direct interaction of **M9** with tubulin has not been demonstrated. Using our fluorescence anisotropy displacement assay, we have thus determined the binding affinity of **M9** for tubulin.

The DM1-lysine conjugate **M9** (Fig. 3c) was prepared from DM1 according to literature procedures[22] (for details, see Supplementary Information). The reaction produced two

high-performance liquid chromatography (HPLC)-separable, but slowly interconverting, C9'-diastereoisomers (designated as **M9A** and **M9B**) in an approximately 1:1 ratio, with full equilibration in phosphate buffer pH 7.2 being observed after 24 h. This finding is in line with previous data suggesting that the thiol addition products of N-ethyl maleimide with different biological thiols (including cysteine and glutathione) readily interconvert even at neutral pH[24]. The method used to prepare **M9** inevitably leads to two diastereoisomeric products, due to the formation of an additional stereocenter at C9' in the 1,4-addition of DM1 to the maleimido part of the linker precursor and the inherently non-selective nature of the reaction. However, the interconversion rate between **M9A** and **M9B** is sufficiently slow to allow the separate assessment of their tubulin-binding affinities. As shown in Fig. 3d, both isomers bind to tubulin with comparable $K_d$ values of 12.6 ± 0.4 nM (mean ± SEM; $n = 3$) and 17.5 ± 1.2 nM (mean ± SEM; $n = 3$) for **M9A** and **M9B**, respectively, values similar to the one obtained for the maytansine derivative ansamitocin P3. These $K_d$ values are in the same ballpark as the one obtained for FcMaytansine using direct titration experiments (see above), indicating that the fluorescein moiety does not significantly influence the affinity of FcMaytansine for tubulin.

Together, these data demonstrate that the affinity of **M9** is very similar to that of S-Me DM1, which suggests that the difference in antiproliferative activity when the compounds are added to cells exogenously is due to differences in cell uptake. In contrast, the potency of intracellularly generated **M9** should be comparable with that of maytansine (and, thus, of S-Me DM1).

**Spongistatin-1 and disorazole Z are maytansine-site ligands.** Based on biochemical experiments, canonical maytansine-site ligands like maytansine (or its derivative ansamitocin P3) and

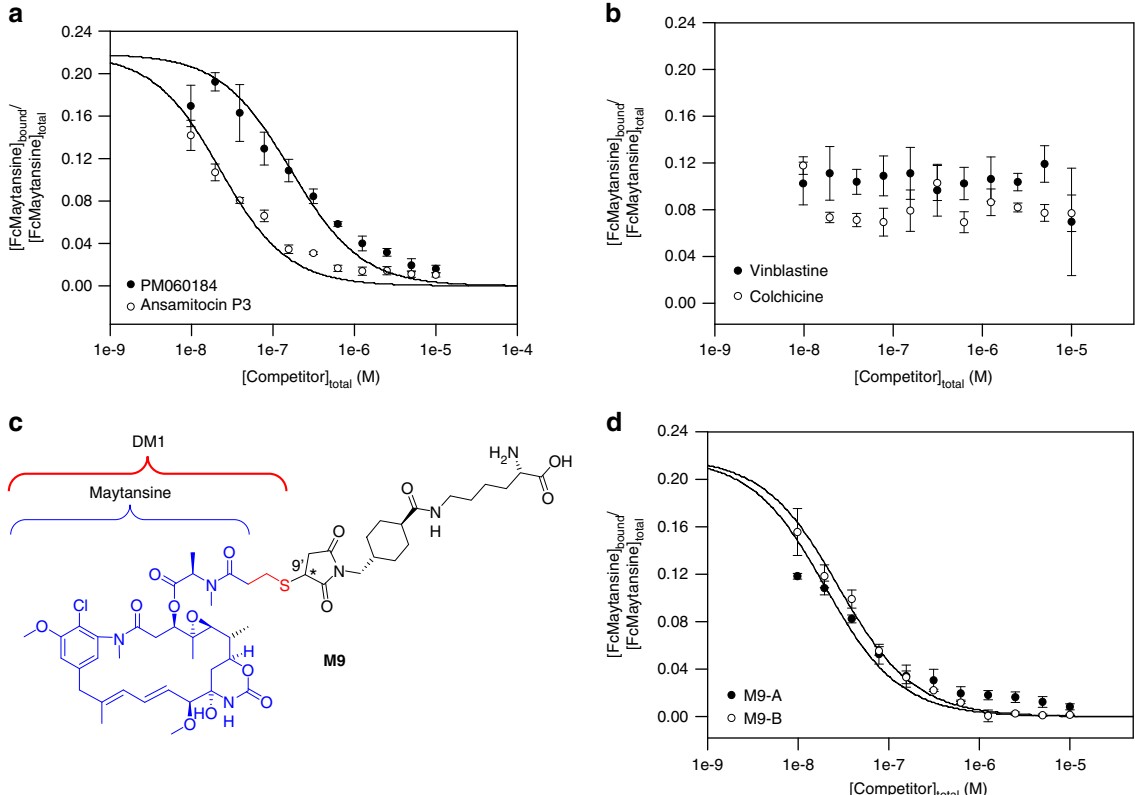

**Fig. 3** Characterization of maytansine-site ligands. **a**, **b** Displacement assays for PM060184 (open circles) and ansamitocin P3 (closed circles) (**a**), and for vinblastine (closed circles) and colchicine (open circles) (**b**). **c** Molecular structure of the trastuzumab emtansine derivative **M9**. **d** Displacement assays for the two diastereoisomers of **M9**, **M9A** (closed circles) and **M9B** (open circles). The data are from three independent experiments and represent mean ± SEM. The solid lines represent fits to the data (see Methods)

rhizoxin had been previously thought to bind to the vinblastine site of tubulin[18,25–27]. This misassignment can be explained by the fact that the maytansine and vinblastine sites are located close to each other and, thus their distinction is not straightforward (Supplementary Fig. 2a). To assess whether our fluorescence anisotropy assay would also enable the identification of additional maytansine site-specific ligands, we tested a small number of additional compounds, including the natural product spongistatin-1. Spongistatin-1 is a highly potent tubulin polymerization inhibitor that is believed to bind to the vinblastine site on β-tubulin[28]. Intriguingly, spongistatin-1 readily displaced FcMaytansine from tubulin in a dose-dependent manner, revealing a $K_d$ of $9.5 \pm 1.9$ nM (mean ± SEM; $n = 3$) (Fig. 4a, b). Likewise, the potent cytotoxin disorazole Z[29], which belongs to the same family of myxobacterial natural products as the vinblastine-site ligand disorazole $C_1$[30], but whose binding to tubulin had not been investigated previously, interacts with the maytansine site with a $K_d$ of $576 \pm 49$ nM (mean ± SEM; $n = 3$) (Fig. 4c, d).

To test whether the two compounds indeed bind to the maytansine site of tubulin, we used our $T_2R$–TTL crystal system and solved the tubulin–spongistatin-1 and tubulin–disorazole Z complex structures to 2.4 and 2.1 Å resolution, respectively (Table 1; Supplementary Fig. 2bc). The two complex structures revealed that both compounds indeed interact with the maytansine site on β-tubulin without inducing any significant conformational changes in the protein (Fig. 4e, f). Spongistatin-1 binds to the maytansine site by establishing a hydrophobic network between its cyclic core and the side chains of βAla397, βMet398, βAla403, βPhe404, βVal181 and βTrp407. Its side chain is buried in a pocket shaped by the side chains of residues

βPro173, βPro175, βGlu183, βPro184 and βGln394, and by the main chain atoms of βSer178, βThr180 and βVal181. This accounts for the structure–activity relationship finding that incorporation of the full side chain is critical for the potent cytotoxicity of spongistatin-1[31]. The tubulin–spongistatin complex is further stabilized by polar contacts formed between the main chain carbonyl group of βAsp179 and the hydroxyl group of the F ring and between the main chain NH group of βVal181 and the hydroxyl group of the B ring of spongistatin. Furthermore, a weak C-H hydrogen bond exists between C28H of the molecule and the main chain carbonyl group of βAla397. The disorazole Z molecule is anchored into the maytansine site by several hydrophobic contacts to residues βTyr408, βTrp407 and βPhe404. The complex is further stabilized by polar contacts between both the oxazole nitrogen and the adjacent carbonyl group of the ester moiety to the main chain NH groups of βVal181 and βVal182, respectively. Moreover, both the carbonyl group of the second ester moiety and the adjacent hydroxyl group form hydrogen bonds to the side chain amide of βAsn101 and the backbone carbonyl of βGly100, respectively. These interactions involve only one half of the disorazole Z molecule. The other half, with both halves being related by twofold symmetry, is exposed to the solvent.

Together, these results demonstrate that our fluorescence anisotropy assay is suitable to identify maytansine-site ligands with $K_d$ values in the nanomolar range. They further establish spongistatin-1 and disorazole Z as maytansine-site ligands.

**Comprehensive description of the maytansine site of tubulin.** Collectively, the crystal structures of tubulin in complex with

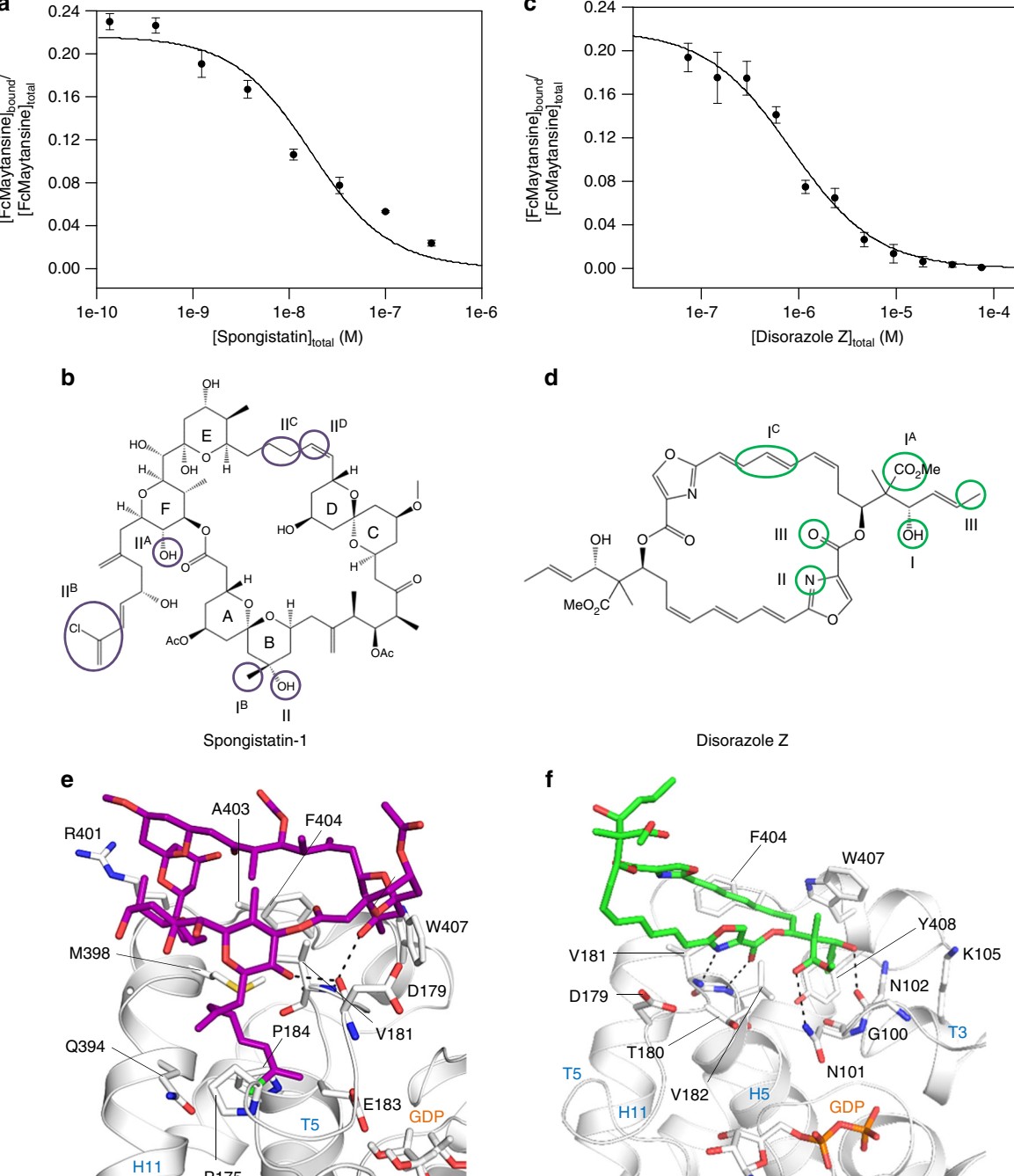

**Fig. 4** Spongistatin-1 and disorazole Z are maytansine-site ligands. **a**, **c** Displacement assays for spongistatin-1 (**a**) and disorazole Z (**c**). The data are from three independent experiments and represent mean ± SEM. The solid lines represent fits to the data (see Methods). **b**, **d** Molecular structures of spongistatin-1 (**b**) and disorazole Z (**d**). The tubulin-interacting parts of the molecules are depicted with purple (spongistatin-1) and green (disorazole Z) circles. The main interacting residues are labeled in black and hydrogen bonds are represented by black broken lines. Secondary structure elements are depicted in blue. **e**, **f** Close-up views of the interaction mode between spongistatin-1 and tubulin (**e**) and between disorazole Z and tubulin (**f**). The β-tubulin subunit is represented in light gray cartoon representation and interacting residues are labeled in black. Spongistatin-1 and disorazole Z are shown in purple and green sticks representation, respectively. Oxygen, nitrogen and sulfur atoms are colored in red, blue and yellow and hydrogen bonds are represented with black broken lines. GDP molecule is shown in orange spheres representation. Secondary structural elements are labeled in blue

spongistatin-1 and disorazole Z together with those of the complexes with maytansine, PM060184, and rhizoxin F[9] allowed us to analyze the interactions of ligands with the maytansine site in significant detail. To this end, we superimposed all five complex structures and compared the binding modes of the structurally distinct ligands (Fig. 5a, b). Maytansine, PM060184 and rhizoxin F have been previously shown to share three key interaction

points with β-tubulin (denoted I, II and III), which define a common pharmacophore[9]. As illustrated in Fig. 5c, points I and II represent hydrogen bond acceptor sites, mediating interactions with βAsn102, βGly100 (point I) and with βVal181 (point II), respectively; point III mediates hydrophobic contacts through βVal182 and βTyr408. Notably, while disorazole Z binds to all three common pharmacophore acceptor points, Spongistatin-1

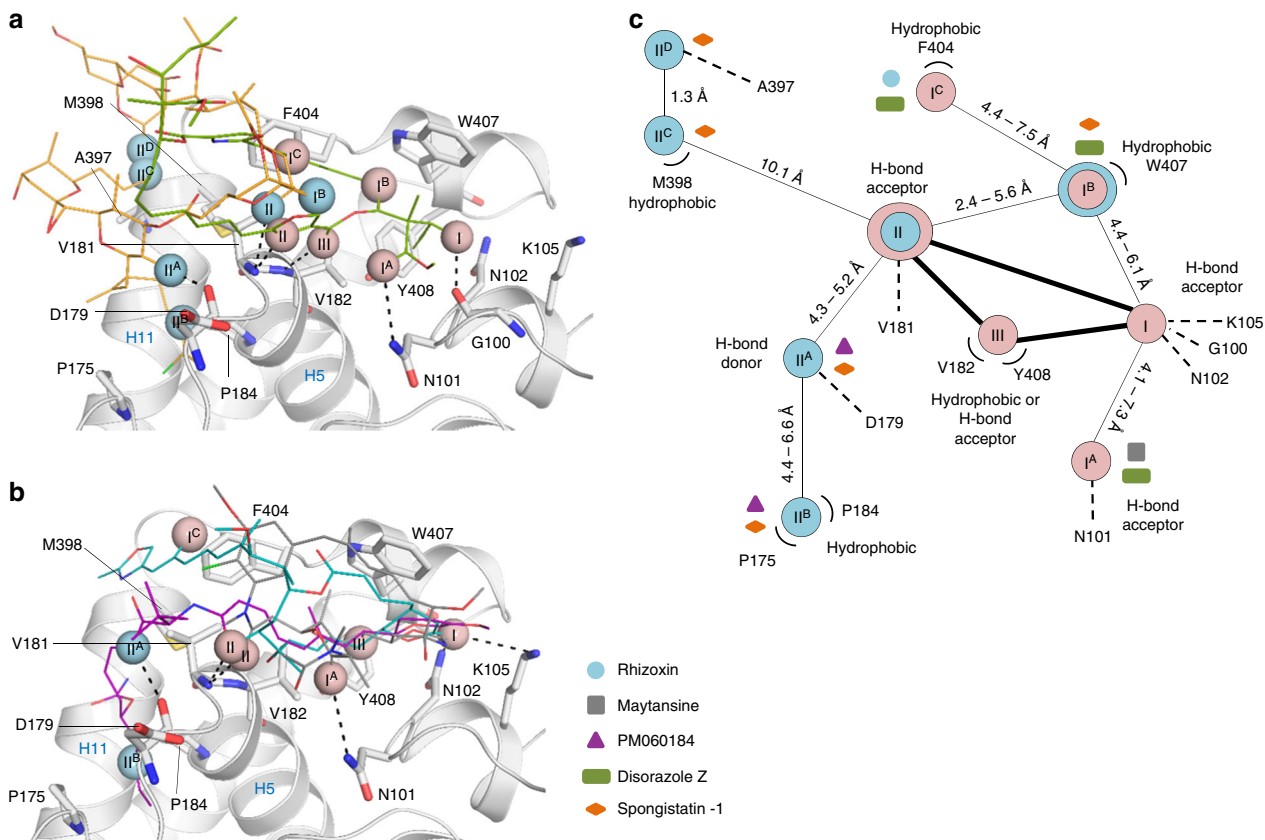

**Fig. 5** Maytansine-site pharmacophore. **a** Superposition of tubulin-bound spongistatin-1 (orange) and disorazole Z (green) (**b**) Superposition of tubulin-bound maytansine (gray), rhizoxin F (cyan) and PM060184 (purple). The main interacting residues are labeled in black and hydrogen bonds are represented by black broken lines. Secondary structure elements are labeled in blue. The interacting parts of the molecules are depicted with spheres and labeled according to **c**. **c** Schematic drawing of the maytansine-site pharmacophore with additionally explored regions. The secondary interaction points are labeled $I^A$, $I^B$ and $I^C$ and $II^A$, $II^B$, $II^C$ and $II^D$. Differently colored symbols are assigned to each of the five maytansine-site ligands, which appear close to the corresponding interacting residues

binds only to point II. However, all five ligands explore additional, partially common, nearby β-tubulin pockets (Fig. 5c). Maytansine/disorazole Z, spongistatin-1/disorazole Z and rhizoxin/disorazole Z interact with three distinct pockets located close to the common pharmacophore point I via a hydrogen bond acceptor point $I^A$ (mediated by βAsn101), a hydrophobic contact point $I^B$ (mediated by βTrp407) and a hydrophobic contact point $I^C$ (mediated by Phe404), respectively. Furthermore, PM060184 and spongistatin-1 both interact with two pockets adjacent to the common pharmacophore point II via hydrogen bonds (denoted point $II^A$; mediated by βAsp179) and hydrophobic contacts (denoted point $II^B$; mediated by βPro175 and βPro184). Interestingly, spongistatin-1 establishes two distinct interactions close to point II via a hydrophobic contact (denoted point $II^C$; mediated by βMet398) and a weak hydrogen bond (point $II^D$; mediated by βAla397), thereby defining an additional binding mode that is not observed for any of the other four maytansine-site ligands. Overall, this analysis provides a comprehensive and detailed comparison of the tubulin-binding modes of maytansine-site ligands.

## Discussion

In this study, we have established a high-performance assay that allows the identification and quantitative evaluation of ligands that specifically bind to the maytansine site of tubulin. Our FcMaytansine probe enables the screening of ligands with $K_d$ values in the nanomolar range in a fast, cheap, non-destructive

and easily automatable manner. Using this assay, we determined the dissociation constants of a number of important maytansine-site ligands, including the pharmacologically active entity of the ADC trastuzumab emtansine and the phase II compound PM060184. We further identified spongistatin-1 and disorazole Z as canonical maytansine-site ligands and we have structurally characterized the maytansine site in great detail.

Our results have several implications. Firstly, our experimental platform will allow the curation of assignments of ligands that may be thought to bind to the vinblastine site erroneously. We expect that the number of ligands binding to the maytansine site will further increase and establish it as a major ligand binding site on tubulin along with the taxane, colchicine and vinblastine sites. Secondly, our FcMaytansine approach paves the way to the generation of additional tool compounds to be used in in vitro and in cellular experiments. The maytansine site of β-tubulin is exposed at the plus-end of microtubules. Exchanging the fluorescein moiety of FcMaytansine by a stronger fluorophore could thus be used to visualize distal microtubule plus-ends by fluorescence microscopy and study the action of single drug molecules on their molecular target. Thirdly, maytansine derivatives bearing fluorescent groups other than fluorescein could be used to develop assays for the identification of low micromolar affinity binders, as they are typically found in small molecule and fragment libraries. To the best of our knowledge, there are no structurally simple (as compared to the structurally complex natural products investigated here), potent maytansine-site lead compounds available that could potentially be developed into

(free) drugs for clinical use. On the other hand, the detailed structural description of the maytansine site may now enable the rational design of such less complex maytansine-site-directed MTAs. Overall, our study presents a possible basis for the development of next-generation anti-tubulin drugs for the treatment of cancer.

## Methods

**Chemistry and compounds.** Details of the synthesis of the fluorescent maytansinoid **M5** and the DM1-lysine conjugate **M9** are reported in the Supplementary Information (see Supplementary Methods and Supplementary Figs 3–6). The synthesis of spongistatin-1 and PM060184 are described elsewhere[32,33]. Disorazol Z was produced by fermentation as described in the corresponding patent literature (biologically active compounds obtainable from Sorangium cellulosum, EP 1743897 A1). Ansamitocin P3, colchicine and vinblastine were obtained from Sigma Aldrich. Maytansinol was purchased from Levena Biopharm, San Diego, USA.

**Tubulin-binding of FcMaytansine and competition experiments.** The binding constant of the fluorescent FcMaytansine probe to tubulin was determined by fluorescence anisotropy titration at 25 °C in black Greiner 384-well, flat-bottom microplates and in a final volume of 50 µL. 10 nM FcMaytansine in assay buffer (15 mM PIPES-KOH, pH 6.8, supplemented with 0.3 mM MgCl₂ and 0.2 mM EDTA) was titrated with increasing amounts of tubulin up to 400 nM. The anisotropy and fluorescence values of the free ($r = 0.016 \pm 0.001$, fluorescence = 1 (arbitrary units)) and bound ($r = 0.245 \pm 0.005$, fluorescence = $3.10 \pm 0.05$ (arbitrary units)) states were determined in the absence and presence of 400 nM tubulin, respectively. The anisotropy was measured using a PHERAstar microplate reader (BMG Labtech) with the fluorescence polarization module. Samples were equilibrated at 25 °C for 15 min prior to measurement at 25 °C. Excitation and emission wavelengths were set to 485 and 528 nm, respectively. The read height and the gain were adjusted automatically. As a canonical maytansine-site compound we used ansamitocin P3, a close derivative of the parent compound maytansine. Comparison of the high-resolution tubulin-ansamitocin P3 crystal structure (our own unpublished results) with that of tubulin–maytansine[9] showed that all tubulin-contacting groups of the two compounds are identical. We thus consider ansamitocin P3 in this context as an excellent substitute for maytansine.

Fluorescence anisotropy competition experiments and analyses were performed as previously described[34] with minor modifications. Samples containing 10 nM FcMaytansine and 4 nM tubulin in assay buffer were titrated with increasing amounts of a competitor at 25 °C in black Greiner 384-well, flat-bottom microplates and in a final volume of 50 µL. Fluorescence anisotropy measurements were carried out as described above.

In order to calculate the binding constant of FcMaytansine and one of a non-fluorescent ligand from a competition experiment, the fractional saturation, νb, of FcMaytansine has to be determined[34]. Binding of FcMaytansine to tubulin was measured through changes in its anisotropy. The anisotropy of a mixture of free and bound FcMaytansine can be expressed as

$$r = Fb * rb + Ff * rf, \tag{1}$$

where $r$ is the measured anisotropy, Ff and Fb are the fractional fluorescence intensities of free and bound FcMaytansine, respectively, rf is the anisotropy of the free FcMaytansine, and rb is the anisotropy of the bound FcMaytansine.

In the simplest case of a fluorescent probe with no quantum yield difference between the bound and the free state[35], the fraction of bound (νb) and free (νf) fluorophore coincides with the fluorescence fractions for the bound and free fluorophore (Fb and Ff, respectively). Employing Eq. 1 and knowing that the sum of the fluorescence fractions is 1 by definition, we obtain

$$\nu b = (r - rf) / (rb - rf). \tag{2}$$

However, FcMaytansine does not exhibit this behavior, because there is a $3.1 \pm 0.1$-fold difference between the fluorescence intensity of the bound and free FcMaytansine. In this case the fluorescence fraction is weighted by the change in fluorescence intensity according to

$$Fb = \nu b * (Ib / It), \tag{3}$$

where Ib is the fluorescence intensity of bound FcMaytansine and It is the total fluorescence intensity. A similar expression applies to free FcMaytasine. Given Eqs. 1 and 3

$$r = (If / It * \nu f * rf) + (Ib / It * \nu b * rb), \tag{4}$$

where If is the fluorescence intensity of free FcMaytansine and vf is the fractional saturation of free FcMaytansine. Since the sum of the fractions free (vf) and bound (vb) ligand is 1 and

$$It = If * \nu f + Ib * \nu b. \tag{5}$$

Employing Eqs. 1, 4 and 5, we obtain the fractional saturation of FcMaytansine needed to calculate the binding constant of its binding to the site and this of a ligand by competition with a probe of known binding constant.

$$\nu b = (r - rf) / [(r - rf) + R(rb - r)], \tag{6}$$

with $R$ representing the ratio between the fluorescence intensity of the bound and free species ($R = Ib/If$). Equation 6 can be simplified with Eq. 3 in cases when there is no change in the quantum yield (i.e., $R = 1$)[36].

Once the fractional saturation of FcMaytansine in the experiments is known, the free concentration of tubulin and FcMaytansine in the titration assay can be calculated from the difference between the bound and the total concentration of the probe. The free concentration of tubulin was calculated from the difference in the total and bound concentration of FcMaytansine assuming a stoichiometry of 1:1. The values of fractional saturation of FcMaytansine bound vs. the free tubulin concentration were used to determine a binding constant at 25 °C of $1.6 \pm 0.3 \times 10^8$ M$^{-1}$ using SigmaPlot v13 (Systat Software, Inc., San Jose, CA, USA).

The binding constant of a problem ligand K(l) can be determined from the known values of the binding constant of the reference ligand K(r) and the total concentrations of binding sites, FcMaytansine and problem ligand by solving the simultaneous mass action equations:

$$K(l) = [Ligand]_{bound} / [Sites]_{free} * [Ligand]_{free} \tag{7}$$

$$K(r) = [FcMaytansine]_{bound} / [Sites]_{free} * [FcMaytansine]_{free} \tag{8}$$

$$[FcMaytansine]_{free} = [FcMaytansine]_{total} - [FcMaytansine]_{bound} \tag{9}$$

$$[Ligand]_{free} = [Ligand]_{total} - [Ligand]_{bound} \tag{10}$$

$$[Sites]_{free} = [Sites]_{free} - [Ligand]_{bound} - [FcMaytansine]_{bound}. \tag{11}$$

The system was solved using EQUIGRA5[34] using the binding constant of FcMaytansine and a stoichiometry of 1:1 for the tubulin–FcMaytansine complex (EQUIGRA5 can be obtained upon request from Fernando Díaz, fer@cib.csic.es).

**X-ray data collection and structure solution.** Crystals of T₂R–TTL were obtained by the sitting-drop vapor-diffusion method as previously described[16,17] and with a protein ratio of 2:1.2:1.2 (tubulin/RB3/TTL). Crystals reached their maximum dimensions within 1 week and were soaked for 1 h to overnight at 20 °C in reservoir solution consisting of 10 % PEG 4k, 16% glycerol, 30 mM MgCl₂, 30 mM CaCl₂ and 100 mM 2-(N-morpholino)ethanesulfonic acid/imidazole pH 6.7 and containing 2 or 5 mM of compound (final DMSO concentration: 10%). Crystals were fished directly from the drop, transferred for a few seconds to a reservoir solution supplemented with 2 or 5 mM of compound and 20% glycerol as cryo-protecting agent and flash-cooled in a nitrogen stream at the beamline. Standard data collections were performed at 1 Å wavelength at 100 K at the beamline X06DA at the Swiss Light Source (Paul Scherrer Institut, Villigen, Switzerland). Standard data processing, structure solution using the difference Fourier method and structure refinement were performed with the programs XDS[37] and Phenix[38,16,17]. All models have good geometry, with small r.m.s. deviations from ideal values for bond lengths and bond angles. MolProbity[39] analysis shows all residues in favored or allowed regions in the Ramachandran plot (T₂R–TTL–disorazole Z: 98.2/1.8%; T₂R–TTL-Spongistatin-1: 97.4/2.6%; T₂R–TTL–FcMaytansine: 97.6/2.4%). Data collection and refinement statistics are given in Table 1. Figures were prepared in PyMol (Schrödinger, LLC).

**Data availability.** Data supporting the findings of this manuscript are available from the corresponding author upon reasonable request. Coordinates of the X-ray crystal structures have been deposited in the RCSB PDB (www.rcsb.org) under accession numbers 6FJF (tubulin-FcMaytansine), 6FII (tubulin-spongistatin-1) and 6FJM (tubulin-disorazole Z).

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

## Acknowledgements

We thank V. Olieric and M. Wang for excellent technical assistance with the collection of X-ray data at beamlines X06DA of the Swiss Light Source (Paul Scherrer Institut, Villigen, Switzerland). We are indebted to Kurt Hauenstein and Dr. Tobias Brütsch for valuable technical advice. We also thank Dr. Bernhard Pfeiffer for support with NMR analyses and the mass spec facility at the Laboratory of Organic Chemistry of the ETH Zürich for HRMS measurements, and the UK National Mass Spectrometry Facility (NMSF) at Swansea University. This work was supported by grants from MINECO/FEDER (BFU2016-75319-R; to J.F.D), the Swiss National Science Foundation (31003A_166608; to M.O.S.) and by the COST Action CM1407 (to J.F.D., K.-H.A. and M.O.S.).

## Author contributions

G.M., A.E.P., D.L.A., J.F.D., R.M., I.P., K.-H.A. and M.O.S. designed the research. G.M., A.E.P., D.L.A., P.B., R.J., and H.I. performed the research. G.M., A.E.P., D.L.A., J.F.D., K.-H.A. and M.O.S analyzed the data. G.M., A.E.P., D.L.A., J.F.D., K.-H.A. and M.O.S wrote the paper with input from the other co-authors. M.O.S. coordinated the project.

## Additional information

**Competing interests:** The authors declare no competing interests.

