## [Peer Review File · Nature Communications]

Reviewers' comments:

Reviewer #1 (Remarks to the Author):

This manuscript describes an experimental tool, i.e., fluorescence anisotropy assay, for characterization of ligands targeting the maytansine-site of tubulin, which is useful for the discovery and development of next generation microtubule-targeting agents (MTAs) for the treatment of cancer. This study has significance in the cancer drug discovery, exploring a recently identified maytansine-binding site in tubulin/microtubule and will attract interest from broad range of cancer researchers dealing with cancer biology, medicinal chemistry, and drug discovery. The chemical syntheses, binding assays, protein crystallography for FcMaytansine-tubulin complex (T2R-TTL-FcMaytansine, 2.4 Å resolution) as well as T2R-TTL-Disorazole Z (1.8 Å resolution) and T2R-TTL-Spongistatin-1 (2.6 Å resolution), binding analysis, K_d determinations, etc., were nicely executed. This study certainly shed light on the key interactions of MTAs targeting the maytansine-site. Accordingly, this work warrants publication in Nature Communications, but with minor revisions addressing the points shown below.

(1) Although the authors repeatedly claim a potential utility of the fluorescence anisotropy assay for HTP applications, this aspect is not clearly presented. If there are some examples of HTP assays using a small library of compounds to identify compounds which have not been known as ligands for the maytansine-site, this claim would have been more persuasive.

The authors wrote in the conclusions, "the detailed structural description of the maytansine-site now enables the rational design of small molecules targeting the maytansine-site". Indeed, this aspect is much more important than the HTP assay capability. Thus, this Reviewer recommends proper revisions on this point in the relevant places in the text.

(2) In Figures 1 and 3, as well as Figure S2 the authors show the binding modes and structures of beta-tubulin-ligand complexes. It would have been helpful for readers if there is an additional graphic, showing the alpha-tubulin unit together with the beta-tubulin unit. The authors indicate that the maytansine-site is at the interface of beta- and alpha-tubulin subunits. If it is indeed the case, isn't there any interaction between the loops of alpha-tubulin with the ligands at all? This point is unclear and thus needs to be addressed.

(3) In the Supporting Information, the HRMS analysis is not shown for M3, which appears to be a literature unknown compound. Even when M3 is not purely isolated, HRMS of M3 by LC-MS can verify the existence of the proposed composition. Thus, this data should be added.

(4) Usually, all protein crystal structures are reported to the Protein Data Base and obtain the PDB code numbers. In this way, there is no doubt about the validity of the crystal structures. However, this is not the case here. This may be up to the Journal's Policy, but Table S1 is insufficient to show the validity of the claimed structures.

Reviewer #2 (Remarks to the Author):

This manuscript described an elegant fluorescence anisotropy method to determine the binding site/affinity of compounds that bind to the maytansine binding site on tubulin. They have demonstrated that maytansine and metabolite derived from antibody-drug conjugate (ADC) trastuzumab emtansine binds at the same site as maytansine and with similar affinity, thus explaining that the high in vitro potency of the ADC is a consequence of the high binding affinity of catabolite that is generated inside the cell. The authors also claim that the methodology will facilitate the determination of the actual site of binding of tubulin-interacting compounds and identification of "maytansine site ligands with K_d values in the nanomolar range.

Enthusiasm for this manuscript is dampened by the following factors:

1. It does not seem that the work described here will be of sufficient interest to others in the community or the wider field.
2. While this is a good methodology paper, the applicability/usefulness of this method is

questionable.

3. The authors state on page 7 that "to the best of our knowledge, M9 has not been demonstrated to bind to tubulin". They fail to refer to the publication by Oroudjev et al., ([Mol. Cancer Ther., 9, 2700-13 (2010)] where the dynamic instability of microtubules upon treatment with the ADC that generates the metabolite M9 (DM1-Lys), a poorly potent compound, is described and shown to be similar to one that generates S-Me DM1, a metabolite with high cellular potency.

4. More importantly, there is no data to suggest that identification of compounds with Kd values in the nM range will lead to the discovery of potent drugs for the "next generation of antibody-drug conjugates". In fact, the compound they identify spongistatin -1 has a Kd (7.4) for displacement of Fc-maytansine, which is lower than that of ansamitocin P-3 (Kd =13.1 nM). However spongistatin-1 (from the published literature) appears to be 50 to 100-fold less potent than ansamitocin P-3 or maytansine, making it unsuitable for use in ADCs. A methodical study of the relationship of in vitro cytotoxicity with Kd for Fc-Maytansine displacement is required before the value of Kd measurements can be gauged. What is the relative in vitro potency of the two compounds described that have vastly different Kd values: spongistatin-1 Kd 7.4 nM and disorazole C1 Kd 1000 nM? Also, what is the Kd value of tubulin binders with similar high potency as maytansine, such as Cryptophycin 52, tubulysin or dolastatin 10?

Other comments:

The structure of DM1 in M9 (and several times in the supporting information) is incorrect. The number of methylene groups (shown in red) should only be 2 not 3.

Reference #15 and 24 are the same.

Reviewer #3 (Remarks to the Author):

I found this paper informative, novel, and well written in presenting new data on known compounds as well as a new fluorescence anisotropic-based assay for compounds binding to the same site on tubulin as maytansine. The assay is a logical extension of fluorescence assays including competitive assays that authors have published previously, and will be used by those in the field interested in this set of potent tubulin-binding drugs.

Since the assay is a major point of the paper, reflected even in the title, there are some points about the description of the assay that would improve the paper. In addition, there are a couple of points about the data presented with new compounds using this assay that could be better described. Here are those points:

1. In Methods, p 13, authors describe the relevant equations for a fluorescence anisotropy assay. Two points could be better described - the terms "Fb" and "Fb* " are both used, but the significance of the " * " is not explained. Also near the bottom of p13, the term " R " is introduced but not explained. Although it is clear to me what is meant, it is not explained explicitly and it would not take much to do so.

2. On p 14 the only explanation of how competitive data were analyzed is to refer the reader to a previous paper describing the program EQUIGRAS. This might be all right if the assay and data analysis were not the main point of the paper. But they are the main points of the paper, so authors should spell this out more rather than just pointing to a reference. If a reader wished to adopt the assay in this current paper, could reader obtain the program and use it without having to find and read another paper, for example?

3. In Figure 3 authors present competition data for two molecules not known to bind at the same site as maytansine, and then present crystallographic data documenting the binding of these compounds to the maytansine site. I wonder if authors think that the crystallographic data provide a rationalization of the poor fits of the competition model to the data, shown in Fig 3A and C? The fits are significantly less good than the fits with 'standard' compounds in Fig 2, e.g., especially in Fig 3C. Do these poorer fits provide information about the binding sites of these compounds as

distinguished from the more 'traditional' compounds?

Reviewer #4 (Remarks to the Author):

The authors generate a fluorescein derivative of Maytansine (FcMaytansine), a new class of microtubule depolymerizing drugs. They measure its affinity for tubulin using fluorescence anisotropy and determine its crystal structure.

They use FcMaytansine to measure the affinities of other compounds that are known to bind to the Maytansine site. They also measure the affinities of a breakdown product of Maytansine-conjugated antibodies and show that its affinity is similar to Maytansine itself.

They then use this assay to show that two natural products (spongistatin-1 and disorazole), previously thought to bind in a different (vinblastine) site actually interact at the Maytansine site. They do an excellent job of confirming this result with crystal structures of these compounds bound to tubulin.

The work is solid and generates a tool that will be very useful in identifying new compounds that target the Maytansine site on tubulin. I particularly like the way the authors have used crystal structures to reinforce their conclusions (and to provide a comprehensive description of how different ligands bind the Maytansine site). I recommend it for publication and have minor textual comments.

Minor Comments

1) The authors could add a line to make it clear why they choose the Maytansine site to target (rather than any of the other sites).

2) The authors mention "isothermal titration fluorescence anisotropy". Do they mean this? What is the isothermal titration?

3) The measured K_d of FcMaytansine is significantly lower than previously measured for Maytansine. The authors state that this is due to different assay conditions. Could it also be the result of the fluorescein? If so can they mention this?

4) Z-factors – could the authors explain what a Z-factor is to help the general reader? It would also be helpful to add a brief explanation of why they used FcMaytansine vs FcMaytansine/tubulin for these measurements.

5) Why do the authors compare the K_ds of M9A, M9B to ansamitocin P3 rather than to Maytansine itself?

6) The authors write: "Together, these data demonstrate that the reduced antiproliferative activity of M9 compared to SMe DM1 when administered to cells exogenously is indeed caused by differences in cellular uptake." They should add a bit more explanation to make this clear. Something like "These data show that the affinity of M9 is very similar to that of SMe DM1, which suggests the difference in antiproliferative activity when the compounds are added to cells exogenously is due to differences in cell uptake".

7) The authors write "Thirdly, the FcMaytansine scaffold could be used as a starting point for the development of probes to screen low micromolar affinity binders.". Could they explain more about what they mean here? What is the benefit of looking for low micromolar affinity binders? How does using a high affinity compound help?

Gramatical suggestions.

1) The following sentence in the abstract doesn't sound quite right "crystal structures of spongistatin-1 and disorazole Z in complex with tubulin allowed defining a novel sub-site". Perhaps "allowed definition of" would be better.

2) The authors write "FcMaytansine is a very specific probe that enables to distinguish between ligands". I would use "can" rather than "enables to".

Reviewer #1:

This manuscript describes an experimental tool, i.e., fluorescence anisotropy assay, for characterization of ligands targeting the maytansine-site of tubulin, which is useful for the discovery and development of next generation microtubule-targeting agents (MTAs) for the treatment of cancer. This study has significance in the cancer drug discovery, exploring a recently identified maytansine-binding site in tubulin/microtubule and will attract interest from broad range of cancer researchers dealing with cancer biology, medicinal chemistry, and drug discovery. The chemical syntheses, binding assays, protein crystallography for FcMaytansine-tubulin complex (T2R-TTL-FcMaytansine, 2.4 Å resolution) as well as T2R-TTL-Disorazole Z (1.8 Å resolution) and T2R-TTL-Spongistatin-1 (2.6 Å resolution), binding analysis, K_d determinations, etc., were nicely executed. This study certainly shed light on the key interactions of MTAs targeting the maytansine-site. Accordingly, this work warrants publication in Nature Communications, but with minor revisions addressing the points shown below.

(1) Although the authors repeatedly claim a potential utility of the fluorescence anisotropy assay for HTP applications, this aspect is not clearly presented. If there are some examples of HTP assays using a small library of compounds to identify compounds which have not been known as ligands for the maytansine-site, this claim would have been more persuasive. The authors wrote in the conclusions, “the detailed structural description of the maytansine-site now enables the rational design of small molecules targeting the maytansine-site”. Indeed, this aspect is much more important than the HTP assay capability. Thus, this Reviewer recommends proper revisions on this point in the relevant places in the text.

> As suggested by this reviewer, we have carefully revised this point in the Abstract, Introduction and Results and Discussion sections of our revised manuscript. In contrast to the HTP capabilities, we now more strongly emphasize that the experimental tools and results presented in our study should contribute to the discovery and characterization of novel maytansine-site-directed, small molecule MTAs for the development of next generation anti-tubulin drugs for the treatment of cancer.

(2) In Figures 1 and 3, as well as Figure S2 the authors show the binding modes and structures of beta-tubulin-ligand complexes. It would have been helpful for readers if there is an additional graphic, showing the alpha-tubulin unit together with the beta-tubulin unit. The authors indicate that the maytansine-site is at the interface of beta- and alpha-tubulin subunits. If it is indeed the case, isn't there any interaction between the loops of alpha-tubulin with the ligands at all? This point is unclear and thus needs to be addressed.

> We apologize for this misunderstanding; maytansine-site ligands bind only to the exposed end of the beta2-tubulin subunit in our T₂R-TTL crystal system. This end corresponds to the plus end of the growing microtubule where the beta-alpha tubulin interface is formed when an incoming tubulin subunit is added. To make this clear, we now show a supplemental figure (Supplementary Fig. 2b) with the overall T₂R-TTL-spongistatin complex structure in ribbon representation (the same result was obtained for disorazole Z).

(3) In the Supporting Information, the HRMS analysis is not shown for M3, which appears to be a literature unknown compound. Even when M3 is not purely isolated, HRMS of M3 by LC-MS can verify the existence of the proposed composition. Thus, this data should be added.

> HRMS data for **M3** have been included in the Supplementary Information of the revised manuscript.

(4) Usually, all protein crystal structures are reported to the Protein Data Base and obtain the PDB code numbers. In this way, there is no doubt about the validity of the crystal structures. However,

this is not the case here. This may be up to the Journal's Policy, but Table S1 is insufficient to show the validity of the claimed structures.

> All the reported structures have been deposited to the RCSB PDB (www.rcsb.org) under accession number 6FJF (tubulin-FcMaytansine), 6FII (tubulin-spongistatin-1) and 6FJM (tubulin-disorazole Z), as stated in the section "ACCESSION NUMBERS" in the original manuscript. Validation reports were also provided together with the submitted manuscript. The coordinates will be released upon publication.

Reviewer #2:

This manuscript described an elegant fluorescence anisotropy method to determine the binding site/affinity of compounds that bind to the maytansine binding site on tubulin. They have demonstrated that maytansine and metabolite derived from antibody-drug conjugate (ADC) trastuzumab emtansine binds at the same site as maytansine and with similar affinity, thus explaining that the high in vitro potency of the ADC is a consequence of the high binding affinity of catabolite that is generated inside the cell. The authors also claim that the methodology will facilitate the determination of the actual site of binding of tubulin-interacting compounds and identification of "maytansine site ligands with K_d values in the nanomolar range. Enthusiasm for this manuscript is dampened by the following factors:

1. It does not seem that the work described here will be of sufficient interest to others in the community or the wider field.

> The experimental tools developed in the course of our study have enabled for the first time to establish spongistatin and disorazole Z as ligands that target the maytansine-site of tubulin. This result not only corrects the long-standing classification of both molecule as vinblastine-site ligands (Bai et al. *Biochemistry* 34, 9714 (1995); Tierno et al. *J. Pharm. Exp. Therap.* 328, 715 (2009)), but it also provides a foundation to significantly expand our understanding of the recently discovered maytansine-site pharmacophore (Figure 4). Furthermore, our method allowed for the first time the characterization of the tubulin-binding of the pharmacologically active metabolite of the clinical antibody-drug conjugate trastuzumab emtansine. Finally, the experimental tools and results presented in our study provide a completely novel and unique framework for the discovery of novel maytansine-site-directed, small molecules for the development of next generation anti-tubulin drugs for the treatment of cancer. Taken together, we thus feel that our study should be of high interest to a broad chemistry, structural biology, biophysical and biochemical readership community.

2. While this is a good methodology paper, the applicability/usefulness of this method is questionable.

> As pointed out in our answer to point 1 of this referee, the applicability/usefulness of our method enabled for the first time to establish two ligand classes as maytansine-site ligands, to significantly expand the maytansine-site pharmacophore, and to investigate the binding properties of the pharmacologically active metabolite of the clinical antibody-drug conjugate trastuzumab emtansine. We also show that the assay is highly specific for the maytansine-site and can be operated in a high-throughput manner, which will allow screening of large compound libraries in a fast, cheap, non-destructive, and easily automatable manner. We feel that altogether these major achievements would not have been possible without the methodology developed in the course of this study.

3. The authors state on page 7 that "to the best of our knowledge, M9 has not been demonstrated to bind to tubulin". They fail to refer to the publication by Oroudjev et al., (*Mol. Cancer Ther.*, 9, 2700-13 (2010)) where the dynamic instability of microtubules upon treatment with the ADC that

generates the metabolite M9 (DM1-Lys), a poorly potent compound, is described and shown to be similar to one that generates S-Me DM1, a metabolite with high cellular potency.

> We thank the reviewer for this comment, which points to a need for some clarification on the novelty of our findings for compound **M9**. The reviewer is correct in that the above paper by Oroudjev et al. has demonstrated that (i) an anti-EpCAM-DM1 conjugate with a non-cleavable linker strongly inhibits the proliferation of MCF-7 cells, that (ii) **M9** is formed from this conjugate intracellularly, and that (iii) the increase in intracellular **M9** concentration over time correlates with the effects of anti-EpCAM-DM1 treatment on microtubule dynamicity. The latter finding indeed "indicate[s] that the metabolites and the free unconjugated maytansinoids have similar mechanisms of action, exerting their antiproliferative effects by inhibiting mitosis through suppression of microtubule dynamic instability" (quote from the Oroudjev paper). However, what Oroudjev et al. did not demonstrate (and did not investigate) is a direct interaction between **M9** and tubulin and this is what our statement is referring to; while persuasive, the evidence from the Oroudjev paper for an interaction between tubulin and **M9** is still indirect and it also leaves open the possibility of different binding sites for maytansine and **M9**. Obviously, the experiments carried out by Oroudjev also did not allow to assess the effects of the two diastereoisomers of **M9** on microtubule dynamics, as both diastereoisomers are produced intracellularly and the compounds were not isolated. Thus, our work also provides the first demonstration that both **M9** diastereoisomers bind to tubulin with equal affinity and, therefore, will exert the same pharmacological activity. We have clarified this point on pages 7 and 8 of the revised version of our manuscript.

4. More importantly, there is no data to suggest that identification of compounds with Kd values in the nM range will lead to the discovery of potent drugs for the "next generation of antibody-drug conjugates". In fact, the compound they identify spongistatin -1 has a Kd (7.4) for displacement of Fc-maytansine, which is lower than that of ansamitocin P-3 (Kd =13.1 nM). However spongistatin-1 (from the published literature) appears to be 50 to 100-fold less potent than ansamitocin P-3 or maytansine, making it unsuitable for use in ADCs. A methodical study of the relationship of in vitro cytotoxicity with Kd for Fc-Maytansine displacement is required before the value of Kd measurements can be gauged. What is the relative in vitro potency of the two compounds described that have vastly different Kd values: spongistatin-1 Kd 7.4 nM and disorazole C1 Kd 1000 nM? Also, what is the Kd value of tubulin binders with similar high potency as maytansine, such as Cryptophycin 52, tubulysin or dolastatin 10?

> Nowhere in the manuscript do we claim that the identification of compounds with Kd values in the nanoM range will lead to the discovery of potent drugs for the next generation of ADCs. In the following, we nevertheless address the reviewer's points concerning the relationship between affinity and cytotoxicity of compounds binding to the same site on tubulin.

The relationship between cytotoxicity and binding affinity has been previously assessed for epothilones (Buey et al., Chem. Biol. 11, 225 (2004)), other taxane-site ligands (Buey et al. Chem. Biol. 12, 1269 (2005)), taxanes (Matezanz et al., Chem. Biol. 15, 573 (2008), and discodermolide/dyctiostatin (Trigili et al., ACS Omega 1, 1192 (2016)). These studies have demonstrated that cytotoxicities and binding affinities correlate very well as long as the binding site is the same and the core of the molecule is kept constant (i.e., only substituents are modified). The reason for this is that the cytotoxicity of a molecule in cells depends on many factors (e.g., susceptibility to metabolic cleavage, differential uptake, solubility, etc.) and not only on the actual binding strength to the target itself. In summary, these studies showed that the determination of the affinity of a ligand for tubulin is an excellent tool for the optimization of a given chemotype drug, but

it is not possible to correlate binding affinity with cytotoxicity if compounds are chemically unrelated, especially if they do not target the same binding site.

Concerning the cytotoxicity values, a study in a panel of solid tumor cell lines showed IC50 values between 30 and 100 pM for maytansine (Widdison et al., *J. Med. Chem.* 49, 4392 (2006)). In contrast to the values indicated by the reviewer, spongistatin-1 has been demonstrated to have single digit picomolar (!) antiproliferative activity against a large panel of human cancer cell lines as well as in vivo antitumour activity in mouse xenograft models (Xu et al., *Anticancer Res.* 31, 2773 (2011); Rothmeier et al., *Int. J. Cancer* 127, 1096 (2010); Paterson et al., *Chem. Commun.* 4, 462 (2003)). In the case of disorazole, the binding affinity is 450 nM (present study), ~100 times higher than that of maytansine and spongistatin. However, IC50 values of disorazoles are in the low nanoM range (Tierno et al., *J. Pharmacol. Exp. Ther.* 328, 715 (2009); Chad et al., *Org. Lett.*, 13, 4088 (2011)). Thus, considering the structural variability between maytansine, spongistatin and disorazole, we find that the ratios between the affinities are well reflected in the ratios of the corresponding cytotoxicities.

Cryptophycin 52, tubulysin and dolastatin 10: Cryptophycin 52 displays a Kd of 47 nM and an IC50 value of 20 pM (Panda et al., *Proc Natl Acad Sci USA* 95, 9313 (1998)); however, its exact binding site on tubulin, to the best of our knowledge, is unknown. Tubulysin and dolastatin 10 bind to the vinca site of tubulin (our unpublished results), which is distinct from the maytansine site and thus are expected to display a completely different molecular mechanism of action. We are not aware of any report on Kd values for the two latter compounds.

In summary, we feel that the above elaborations and discussions, although interesting, are not relevant to the main line of our study. We thus decided not to include any of these points in the revised version of our manuscript.

Other comments:

The structure of DM1 in M9 (and several times in the supporting information) is incorrect. The number of methylene groups (shown in red) should only be 2 not 3.

> We are grateful to the reviewer for having pointed this out. All corresponding structures have been corrected in the revised version of our manuscript.

Reference #15 and 24 are the same.

> We have corrected this mistake

Reviewer #3:

I found this paper informative, novel, and well written in presenting new data on known compounds as well as a new fluorescence anisotropic-based assay for compounds binding to the same site on tubulin as maytansine. The assay is a logical extension of fluorescence assays including competitive assays that authors have published previously, and will be used by those in the field interested in this set of potent tubulin-binding drugs.

Since the assay is a major point of the paper, reflected even in the title, there are some points about the description of the assay that would improve the paper. In addition, there are a couple of point about the data presented with new compounds using this assay that could be better described. Here are those points:

1. In Methods, p 13, authors describe the relevant equations for a fluorescence anisotropy assay. Two points could be better described - the terms "Fb" and "Fb* " are both used, but the significance of the " * " is not explained. Also near the bottom of p13, the term " R " is introduced but not

explained. Although it is clear to me what is meant, it is not explained explicitly and it would not take much to do so.

> We used the "*" as a multiplication sign, however, did not introduce spaces before and after the symbol. We have now consistently changed this (also for the division and equal signs) for all formulae presented throughout the revised version of our manuscript. The term "R" corresponds to the ratio of the fluorescence intensities of the bound and the free forms. This is now explained explicitly in the corresponding Methods section of the revised manuscript.

2. On p 14 the only explanation of how competitive data were analyzed is to refer the reader to a previous paper describing the program EQUIGRAS. This might be all right if the assay and data analysis were not the main point of the paper. But they are the main points of the paper, so authors should spell this out more rather than just pointing to a reference. If a reader wished to adopt the assay in this current paper, could reader obtain the program and use it without having to find and read another paper, for example?

> EQUIGRAS is a software that solves the mass law simultaneous equilibrium of a competition between two ligands. The underlying mathematics is now explicitly described in the revised version of the corresponding Materials and Methods section of our manuscript. The program can be obtained upon request from the authors (now also explicitly mentioned) and is described in Diaz and Buey (Methods Mol. Med. 137, 245 (2007)).

3. In Figure 3 authors present competition data for two molecules not known to bind at the same site as maytansine, and then present crystallographic data documenting the binding of these compounds to the maytansine site. I wonder if authors think that the crystallographic data provide a rationalization of the poor fits of the competition model to the data, shown in Fig 3A and C? The fits are significantly less good than the fits with 'standard' compounds in Fig 2, e.g., especially in Fig 3C. Do these poorer fits provide information about the binding des of these compounds as distinguished from the more 'traditional' compounds?

> We agree with the reviewer that the fits to the data obtained with spongistatin and disorazole Z were significantly less good than those with our reference compounds. This prompted us to repeat these measurements. As shown in the revised Figures 3ab, we have now obtained data that could be fitted in a much more descent manner. In this context, our structural data are strongly suggestive that spongistatin and disorazole Z should indeed behave very similar to our reference compounds. Thus a competitive model seems to be warranted to fit the data obtained with both ligands.

Reviewer #4:

The authors generate a fluoresceine derivative of Maytansine (FcMaytansine), a new class of microtubule depolymerizing drugs. They measure its affinity for tubulin using fluorescence anisotropy and determine its crystal structure.

They use FcMaytansine to measure the affinities of other compounds that are known to bind to the Maytansine site. They also measure the affinities of a breakdown product of Maytansine-conjugated antibodies and show that its affinity is similar to Maytansine itself.

They then use this assay to show that two natural products (spongistatin-1 and disorazole), previously thought to bind in a different (vinblastine) site actually interact at the Maytansine site. They do an excellent job of confirming this result with crystal structures of these compounds bound to tubulin.

The work is solid and generates a tool that will be very useful in identifying new compounds that target the Maytansine site on tubulin. I particularly like the way the authors have used crystal

structures to reinforce their conclusions (and to provide a comprehensive description of how different ligands bind the Maytansine site). I recommend it for publication and have minor textual comments.

Minor Comments

1) The authors could add a line to make it clear why they choose the Maytansine site to target (rather than any of the other sites).

> We have chosen the maytansine-site because it is the least well characterized tubulin-ligand binding site up to now and because no tools are available to characterize the binding of maytansine-site ligands in detail. We have added this sentence in the Introduction section of our revised manuscript.

2) The authors mention “isothermal titration fluorescence anisotropy”. Do they mean this? What is the isothermal titration?

> “Isothermal titration” means that the binding experiment was performed at a constant temperature. We realize that this terminology, while standard in the field of calorimetry (cf isothermal titration calorimetry), might be rather confusing in the context of fluorescence anisotropy binding experiment. We thus decided to remove the term "isothermal titration" in the revised version of our manuscript.

3) The measured K_d of FcMaytansine is significantly lower than previously measured for Maytansine. The authors state that this is due to different assay conditions. Could it also be the result of the fluorescein? If so can they mention this?

> We initially indeed considered the possibility that the fluorescein moiety would influence the affinity of FcMaytansine for tubulin. However, the K_d values obtained for ansamitocin P3, M9A and M9B, three ligands that are very similar to the parent maytansine molecule, are very much comparable to the K_d of FcMaytansine. Based on this result, we concluded that the fluorescein does not strongly contribute to the affinity of FcMaytansine for tubulin.

It should be noted that the K_d determination reported in Lopus et al. (Mol Cancer Ther 9, 2689 (2010)) for maytansine was obtained in a buffer system containing 100 mM PIPES. In our hands, in such high PIPES concentrations tubulin readily forms higher order oligomers (our own unpublished observation). Since maytansine binds to beta-tubulin at the interdimer interface, the presence of oligomers will result in a lower apparent affinity of the ligand due to a competition mechanism. This is the reason why we used 15 mM PIPES, a condition that does not promote tubulin oligomerization.

We now mention on page 6 of the revised manuscript that we think that the fluorescein moiety does not significantly influence the affinity of FcMaytansine for tubulin.

4) Z-factors – could the authors explain what a Z-factor is to help the general reader? It would also be helpful to add a brief explanation of why they used FcMaytansine vs FcMaytansine/tubulin for these measurements.

> The Z-factor is a measure of the statistical effect size and it reports on an assay's signal dynamic range and the data variation associated with the signal measurements. It has been proposed for use in the context of high-throughput screening to judge whether the response in a particular assay is sufficient to warrant further attention (see, for example, (Zhang et al., J. Biomol. Screen. 4, 67 (1999)). We now mention this in the revised Supplementary Information (see legend of Figure S1d).

5) Why do the authors compare the Kds of M9A, M9B to ansamitocin P3 rather than to Maytansine itself?

> Ansamitocin P3 is a derivative of maytansine and is very similar to the parent ligand. We had solved the crystal structure of the tubulin-ansamitocin P3 complex to high resolution before starting this project (unpublished result), and found that all tubulin contacting atoms of the ligand are the same as the ones seen in the tubulin-maytansine structure (Prota et al., Proc. Natl. Acad. Sci. USA 111, 13817 (2014)). We thus consider ansamitocin P3 in this context as an excellent substitute for maytansine. Furthermore and on a more practical side, ansamitocin P3 is much cheaper than maytansine and was readily available to us. We now document this on page 12 of the revised manuscript.

6) The authors write: "Together, these data demonstrate that the reduced antiproliferative activity of M9 compared to SMe DM1 when administered to cells exogenously is indeed caused by differences in cellular uptake." They should add a bit more explanation to make this clear. Something like "These data show that the affinity of M9 is very similar to that of SMe DM1, which suggests the difference in antiproliferative activity when the compounds are added to cells exogenously is due to differences in cell uptake".

> We have changed the sentence accordingly.

7) The authors write "Thirdly, the FcMaytansine scaffold could be used as a starting point for the development of probes to screen low micromolar affinity binders.". Could they explain more about what they mean here? What is the benefit of looking for low micromolar affinity binders? How does using a high affinity compound help?

> To the best of our knowledge, there are no potent maytansine-site lead compounds reported in the literature that are not complex natural products (liked the compounds that were the subject of our study), and could thus be developed into (free) drugs for clinical applications in a straightforward manner. In this context, the FcMaytansine probe could be used as a starting point to develop assays for the identification of low micromolar affinity binders typically present in small molecule and fragment libraries. We now mention this at the end of the Discussion section of our revised manuscript.

Gramatical suggestions.

1) The following sentence in the abstract doesn't sound quite right "crystal structures of spongistatin-1 and disorazole Z in complex with tubulin allowed defining a novel sub-site". Perhaps "allowed definition of" would be better.

> We have changed the wording in the Abstract accordingly.

2) The authors write "FcMaytansine is a very specific probe that enables to distinguish between ligands". I would use "can" rather than "enables to".

> It is not the FcMaytansine probe that makes the distinction, but enables distinction. We thus preferred not to change the wording.

REVIEWERS' COMMENTS:

Reviewer #2 (Remarks to the Author):

This reviewer's comment have been adequately addressed in the revised manuscript

Reviewer #3 (Remarks to the Author):

The authors have successfully addressed the points raised in the first review.

Reviewer #4 (Remarks to the Author):

The authors have thoroughly address my comments and those of the other referees. I strongly recommend publication.

EDITORIAL REQUESTS:

* *Nature Communications* uses a transparent peer review system, where for manuscripts submitted from January 2016 we are publishing the reviewer comments to the authors and author rebuttal letters of our research articles online as a supplementary peer review file. Please let us know in the cover letter when submitting the final version of your manuscript if you wish to opt out of this scheme or not. If you are concerned about the release of confidential data, we do permit redactions in the interest of confidentiality. If you would like to remove such information from these files, then please let us know specifically what information you would like to have removed. Please note that we cannot incorporate redactions for other reasons. For more information, please refer to our FAQ page at <https://media.nature.com/full/nature-assets/ncomms/authors/ncomms-transparent-peer-review.pdf>

> I am happy to publish the reviewer's comments along with the paper. No need from my side to edit them.

* Your manuscript should comply with our policies and format requirements, detailed in our checklist for authors at:

http://www.nature.com/article-assets/npg/ncomms/authors/ncomms_manuscript_checklist.pdf

> I went through the checklist and confirm that the manuscript should formatted accordingly.

* Data availability statements and data citations policy: All *Nature Communications* manuscripts must include a section "Data Availability" at the end of the Methods section or main text (if no Methods). For more information on this policy, and a list of examples, please see <http://www.nature.com/authors/policies/data/data-availability-statements-data-citations.pdf>

- Accession codes for deposited data
- Other unique identifiers (such as DOIs and hyperlinks for any other datasets)
- At a minimum, a statement confirming that all relevant data are available from the authors
- If applicable, a statement regarding data available with restrictions
- If a dataset has a Digital Object Identifier (DOI) as its unique identifier, we strongly encourage including this in the Reference list and citing the dataset in the Data Availability Statement.

> A Data Availability statement is included at the end of the Methods section.

* DATA SOURCES: We strongly encourage authors to deposit all new data associated with the paper in a persistent repository where they can be freely and enduringly accessed. We recommend submitting the data to discipline-specific, community-recognized repositories, where possible and a list of recommended repositories is provided here: <http://www.nature.com/sdata/policies/repositories>

If a community resource is unavailable, data can be submitted to generalist repositories such as figshare (<https://figshare.com/>) or Dryad Digital Repository (<http://datadryad.org/>). Please provide a unique identifier for the data (for example a DOI or a permanent URL) in the data availability statement, if possible. If the repository does not provide identifiers, we encourage authors to supply the search terms that will return the data. For data that have been obtained from publically available sources, please provide a URL and the specific data product name in the data availability statement. Data with a DOI should be further cited in the methods reference section.

> Coordinates of the X-ray crystal structures have been deposited in the RCSB PDB (www.rcsb.org) under accession number 6FJF (tubulin-FcMaytansine; <http://dx.doi.org/10.2210/pdb6FJF/pdb>), 6FII (tubulin-spongistatin-1; <http://dx.doi.org/10.2210/pdb6FII/pdb>) and 6FJM (tubulin-disorazole Z; <http://dx.doi.org/10.2210/pdb6FJM/pdb>).

* To ensure correct hyperlinking of the accession codes in your manuscript, please add the hyperlink or DOI in square brackets directly after the code throughout (for example, '5XRN [<http://dx.doi.org/10.2210/pdb5XRN/pdb>]', '1483958 [<https://dx.doi.org/10.5517/ccdc.csd.cc1t5m6>]', 'SRP109982 [<https://www.ncbi.nlm.nih.gov/sra/?term=SRP109982>]' or 'NQLW00000000 [https://www.ncbi.nlm.nih.gov/assembly/GCA_002312845.1/]').

> We now have included the DOI link of all our PDB deposited structures (see also previous point).

* Please check whether your manuscript or Supplementary Information contain third-party images, such as figures from the literature, stock photos, clip art or commercial satellite and map data. We strongly discourage the use or adaptation of previously published images, but if this is unavoidable, please request the necessary rights documentation to re-use such material from the relevant copyright holders and return this to us when you submit your revised manuscript.

> The SI does not contain any third-party images.

* Nature journals require authors of life sciences research papers to include relevant details about several elements of experimental and analytical design in their manuscripts. This initiative aims to improve the transparency of reporting and the reproducibility of published results and is described at: <http://www.nature.com/authors/policies/reporting.pdf> To ensure that your manuscript complies with our policy, please pay close attention to the 'methods' and 'legends' sections of our checklist for authors: http://www.nature.com/article-assets/npg/ncomms/authors/ncomms_lifesciences_checklist.pdf You may also find the following collection of articles on statistics for biologists helpful: <http://www.nature.com/collections/qghhqm> Additionally, please ensure that an updated editorial policy checklist that verifies compliance with all required editorial policies is completed and uploaded as a Related Manuscript file type with the revised article. Please note that this form is a dynamic 'smart pdf' and must therefore be downloaded and completed in Adobe Reader. <https://www.nature.com/authors/policies/Policy.pdf>

> Done and the corresponding documents have been submitted together with the final materials.

* Please supply the main manuscript file in Microsoft Word or LaTeX format.

> The main manuscript has been written with Microsoft Word.

* The last paragraph of the Introduction which describes the major results and conclusions of the current work should be summarised in present tense.

> Done

* Please provide a discussion section in the main manuscript file.

* We do not permit a Conclusions section. Please rename to 'Discussion' or incorporate the text into the 'Discussion' section, as applicable.

> Done

* Please provide a full Methods section in the main manuscript file. Please note that there are no word limits to the Methods section. The Methods section should contain subheadings that contain fewer than 60 characters including spaces. Please also ensure that these subheadings contain no punctuation.

> A full Methods section is provided after the Discussion section.

* Please remove phrases such as 'new', 'novel', 'for the first time', 'unprecedented', etc. as these are not needed to emphasise the importance of your work.

> All these words have been removed.

* In the Methods, please provide sufficient information such that the experiments could reasonably be reproduced without reference to other papers, and avoid use of the term 'as described previously'.

> We do not use the term 'as described previously' in the final version of the manuscript.

* We are committed to ensuring clarity and avoiding ambiguity in the mathematics in our papers. Consequently, please carefully check the mathematical terms throughout your manuscript and Supplementary Information (including labels on figures and figure captions) to ensure that it conforms strictly to the following guidelines. Equations should be supplied in editable format, and not as images. In mathematical terms, scalar variables (e.g. x , V , χ) should be typeset in italic, whereas multi-letter variables should be formatted without italic. Constants (e.g. \hbar , G , c) should be typeset in italics (the only exceptions being e , i , π , which should be typeset without italic) and vectors (such as r , the wavevector k , or the magnetic field vector B) should be typeset in bold without italics. In contrast, subscripts and superscripts should only be italicized if they too are variables or constants. Those that are labels (such as the 'c' in the critical temperature, T_c , the 'F' in the Fermi energy, E_F , or the 'crit' in the critical current, I_{crit}) should be typeset in roman. Please also ensure the same convention is followed in figure labels, axes, and such. Additionally, to avoid doubt, unit dimensions should be expressed using negative integers (e.g. $\text{kg m}^{-1} \text{s}^{-2}$ not kg/ms^2) or the word 'per'.

> Done

* Please see our requirements regarding characterization of structurally-novel chemical compounds, and the required format for compound characterization data: <http://www.nature.com/ncomms/journal-policies/editorial-publishing-policies#Characterization-materials> Please note that this includes 1H-NMR, 13C-NMR and high resolution mass spectrometry for all structurally-novel chemical compounds.

> Not applicable

* Chemical structures that appear in figures in the manuscript (or as part of the single .cdx file mentioned below), should be drawn using the Nature Chemistry template (available at http://www.nature.com/authors/guides/NR_chemdraw_stylesheet.cds) or using the settings from this template. Structures should be scaled in proportion to fit our figure dimensions, however, please only scale atom labels and bond lengths; please do not reduce (or increase) the bond thickness. Please also refer to the Nature Research Chemical Structures Guide (<https://www.nature.com/authors/guides/ChemStructureGuide.pdf>) to ensure that you prepare your figures in a format that will require minimal changes by our Production teams. Please supply any ChemDraw (.cdx) files with the final version of your manuscript.

> Done

* Please include a stereo image of a portion of the electron density map (including contour level and type of map) for crystallographic structures or the superimposed lowest energy structures (>10) for NMR structures. Please also use the Nature templates for NMR and X-ray refinement statistics (see <http://www.nature.com/ncomms/journal-policies/editorial-publishing-policies#Characterization-materials>). This should be presented as a table in the main manuscript. Please ensure that structural data is deposited in the relevant publicly accessible database, and that accession codes are provided in the Data Availability Statement.

> We included a representative stereo image of a portion of the electron density map (including contour level and type of map) for one of the crystallographic structures (see new Supplemental Fig. 2c). We also did reformat our X-ray crystallography table and included it in the main manuscript as Table 1. As mentioned already above, coordinates of the X-ray crystal structures have been deposited in the RCSB PDB (www.rcsb.org) under accession number 6FJF (tubulin-FcMaytansine; <http://dx.doi.org/10.2210/pdb6FJF/pdb>), 6FII (tubulin-spongistatin-1; <http://dx.doi.org/10.2210/pdb6FII/pdb>) and 6FJM (tubulin-disorazole Z; <http://dx.doi.org/10.2210/pdb6FJM/pdb>); see also Data Availability section of the final manuscript.

* In each Figure and Supplementary Figure where error bars are used, they must be defined, and the number of experimental replicates stated. One statement at

the end of each figure is sufficient if the error bars are equivalent throughout the figure.

> This information is provided for all concerned figure panels.

* Please note that schemes are not permitted and should be re-labelled as figures. Please provide a figure title and a figure caption that is up to 350 words. Please ensure that all figures are re-numbered and cited correctly in the text.

> We have re-labeled our Scheme as Figure 1 and ensured that all figures are re-numbered and cited correctly in the text. We also moved the X-ray crystallography table from the SI to the main manuscript (Table 1).

* Please supply an author contribution section after the acknowledgement section that refers to all authors.

> An Author Contribution section is provided after the Acknowledgments.

* Please make a statement of competing financial and non-financial interests after the author contributions section that refers to all authors. If there are no competing interests, please add the statement "The authors declare no competing interests."

> A corresponding statement is placed after the Author Contributions.

* Please note that we do not reformat Supplementary Information files; they will be uploaded with the published article as they are submitted with the final version of your manuscript. Please check everything very carefully and remove any track changes from the file. Failure to adhere to our style guidelines will result in delays in production. The only sections we permit in the Supplementary Information file are Supplementary Figures, Supplementary Tables, Supplementary Methods, Supplementary Notes, Supplementary Discussion, Supplementary References.

> Done

* In the Supplementary Information file, please ensure that supplementary items are labelled and cited using only the following formats: Supplementary Figure 1, Supplementary Table 1, Supplementary Methods, Supplementary Note 1, Supplementary Discussion, Supplementary References. Please note the use of 'Supplementary' and that we do not use the 'S' prefix.

> Done

* Please replace general citations to the Supplementary Information (e.g. 'see Supplementary Information') with specific citations (e.g. 'See Supplementary Figure 1/Supplementary Table 1/Supplementary Methods/etc.').

> Done

* Each Supplementary Figure should be accompanied by a legend, which should be presented below the figure and may be up to 350 words, that refers to all panels within the figure, and a title that summarises the figure and does not refer to specific panels. This also applies to spectra, which should be labelled as Supplementary Figures.

> Done

* Please ensure that the Supplementary References appear at the end of the SI, and are self-contained and numbered from 1. References mentioned in both the main text and the Supplementary Information should be part of both reference lists so that the Supplementary Information does not refer to the reference list in the main paper and vice versa.

> Done

* Your paper will be accompanied by a two-sentence editor's summary, of between 250-300 characters, when it is published on our homepage. Could you please approve the draft summary below or provide us with a suitably edited version.

EDITOR'S SUMMARY:

Microtubule-targeting agents are used successfully as anticancer therapeutics. Here authors develop a fluorescence anisotropy-based assay to identify and characterize ligands for the maytansine-site of tubulin and provide crystal structures of identified ligands in complex with tubulin.

> Fine as is

OPEN ACCESS:

Nature Communications is a fully open access journal. Articles are made freely accessible on publication under a CC BY license (Creative Commons Attribution 4.0

International License). This license allows maximum dissemination and re-use of open access materials and is preferred by many research funding bodies.

For further information about article processing charges, open access funding, and advice and support from Nature Research, please visit <http://www.nature.com/ncomms/about/open-access>

SUBMISSION INFORMATION:

In order to accept your paper, we require the following:

* A cover letter describing your response to our editorial requests.

> Done

* The final version of your text as a Word or TeX/LaTeX file, with any tables prepared using the Table menu in Word or the table environment in TeX/LaTeX and using the 'track changes' feature in Word.

> Done

* Production-quality versions of all figures, supplied as separate files. To ensure the swift processing of your paper please provide the highest quality, vector format, versions of your images (.ai, .eps, .psd) where available. Please see our brief guide to manuscript submission for further details on the figure formats we can accept. Text and labelling should be in a separate layer to enable editing during the production process. If vector files are not available then please supply the figures in whichever format they were compiled in and not saved as flat .jpeg or .TIFF files. Any chemical structures or schemes contained within figures should additionally be supplied as separate ChemDraw (.cdx) files. If your artwork contains any photographic images, please ensure these are at least 300 dpi.

> We provide each figure as a separate high resolution pdf file.

To ensure that your figures are accessible to colour-blind readers, please consider using alternative colour schemes. For example, rainbow colour scales may be replaced by single-colour intensity scales or greyscale, and red/green image overlays may be replaced with magenta/green. For reference an example of R-script colour blindness palettes can be found here <https://cran.r-project.org/web/packages/viridis/vignettes/intro-to-viridis.html>. Another example for Python can be found here: <http://matplotlib.org/cmoccean/>

> Done

* The final version of any Supplementary Information (figures, tables, notes etc) in one PDF file. Please add a cover page to the Supplementary Information PDF, including the title of the manuscript and the first author's surname in the format 'Smith et al.' Please submit movies, audio files and data sets as separate files. See <http://www.nature.com/ncomms/submit/how-to-submit#Supplementary-information> for acceptable file formats/sizes.

** Please note that Supplementary Information must be finalised prior to acceptance of the paper. **

> Done

* If you wish, an interesting image (but not an illustration or schematic) for consideration as a 'Featured Image' on the Nature Communications homepage. Examples can be seen on our Facebook page:<http://go.nature.com/PGPizM> The file should be 1400x400 pixels in RGB format and should be uploaded as 'Related Manuscript File'. In addition to our home page, we may also use this image (with credit) in other journal-specific promotional material.

> No interest

* A completed author checklist, uploaded as a Related Manuscript file type, available at: http://www.nature.com/article-assets/npg/ncomms/authors/ncomms_manuscript_checklist.pdf

> Done

* Completed and signed copies of our Multimedia License to Publish (LTP) for any Featured Image suggestions (please use one form for each image and give a scientific description of the image in the 'title' field; do not use "Featured Image" as a title): Multimedia Licence to Publish form

> Not applicable

REVIEWERS' COMMENTS:

Reviewer #2 (Remarks to the Author):

This reviewer's comment have been adequately addressed in the revised manuscript

Reviewer #3 (Remarks to the Author):

The authors have successfully addressed the points raised in the first review.

Reviewer #4 (Remarks to the Author):

The authors have thoroughly address my comments and those of the other referees. I strongly recommend publication.